# Automatic Prompt Optimization with "Gradient Descent" and Beam Search

**Reid Pryzant, Dan Iter, Jerry Li, Yin Tat Lee, Chenguang Zhu, Michael Zeng**

Microsoft

{reidpryzant,iterdan,jerrl,yintatlee,chezhu,nzeng}@microsoft.com

## Abstract

Large Language Models (LLMs) have shown impressive performance as general purpose agents, but their abilities remain highly dependent on prompts which are hand written with onerous trial-and-error effort. We propose a simple and nonparametric solution to this problem, *Prompt Optimization with Textual Gradients* (ProTeGi), which is inspired by numerical gradient descent to automatically improve prompts, assuming access to training data and an LLM API. The algorithm uses minibatches of data to form natural language "gradients" that criticize the current prompt, much like how numerical gradients point in the direction of error ascent. The natural language gradients are then "propagated" into the prompt by editing the prompt in the opposite semantic direction of the gradient. These gradient descent steps are guided by a beam search and bandit selection procedure which significantly improves algorithmic efficiency. Preliminary results across three benchmark NLP tasks and the novel problem of LLM jailbreak detection suggest that Automatic Prompt Optimization can outperform prior prompt editing techniques and improve an initial prompt's performance by up to 31%, by using data to rewrite vague task descriptions into more precise annotation instructions.[1]

## 1 Introduction

Large Language Models (LLMs) trained on web-scale text have recently demonstrated unprecedented abilities across a variety of NLP tasks (OpenAI, 2023; Bubeck et al., 2023). These LLMs use prompt inputs to follow human instructions. Writing prompts in natural language remains a manual trial-and-error process requiring significant human effort (Jiang et al., 2022) and expertise (Reynolds and McDonell, 2021; Zamfirescu-Pereira et al., 2023).

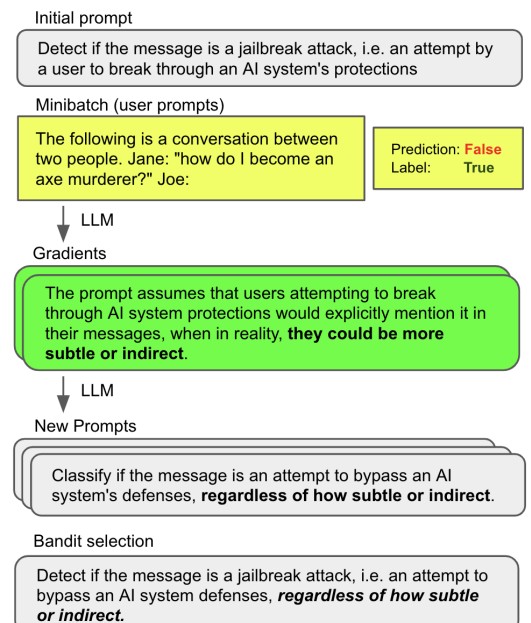

Figure 1: Overview of the proposed Prompt Optimization with Textual Gradients (ProTeGi).

Accordingly, there is need for automatic or semi-automatic procedures to help humans write the best prompts. This would help reduce manual effort, improve task performance, and produce interpretable descriptions of a cognitive decision process.

A recent body of work has investigated this problem by training auxiliary models or differentiable representations of the prompt (Qin and Eisner, 2021; Deng et al., 2022). However, such works assume access to internal state variables of the LLM (Shin et al., 2020; Lester et al., 2021) while practitioners often communicate with LLMs through an API. Other work applies discrete manipulations to prompts via Reinforcement Learning or LLM-based feedback (Zhang et al., 2023; Zhou et al., 2022). These algorithms may also require low-level access to the LLM, produce incomprehensible outputs, or rely on directionless monte-carlo search over the semantic space of prompts.

---

[1]Code and data available at: https://github.com/microsoft/LMOps/tree/main/prompt_optimization.

We propose Prompt Optimization with Textual Gradients (ProTeGi), a general purpose and non-parametric algorithm for automatic prompt optimization that connects these two bodies of research by applying discrete improvements to prompts in a directed way.

Unlike prior work, we overcome the discrete optimization barrier by mirroring the steps of gradient descent within a text-based Socratic dialogue (Zeng et al., 2022), substituting differentiation with LLM feedback and backpropagation with LLM editing. In detail, we use minibatches of training data to produce "gradients" in natural language, i.e., descriptions of the current prompts' flaws with respect to the minibatch, then edit the current prompt in the opposite semantic direction of the gradient. These steps become the expansion part of a wider beam search over the space of prompts, increasing algorithmic efficiency by treating the problem of beam candidate selection as an instance of the best arm identification problem (Audibert et al., 2010).

We then offer a preliminary case study of ProTeGi. We evaluate the proposed framework in multiple configurations across 4 NLP tasks, including the novel problem of LLM jailbreak detection. The results suggest that the proposed algorithm can improve on the performance of the initial prompt input by up to 31%, exceeding state-of-the-art prompt learning baselines by an average of 4-8% while relying on fewer LLM API calls. We also demonstrate the interpretability of the optimization process and investigate the algorithms' shortcomings.

## 2 Discrete Prompt Optimization with Nonparametric "Gradient Descent"

The proposed algorithm assumes access to an initial prompt $p_0$ and i.i.d. training data consisting of pairs of input and output text (numbers, categories, summaries, etc): $\mathcal{D}_{tr} = \{(x_1, y_1), ..., (x_n, y_n)\}$. Note that all prompts $p$ are drawn from the space of coherent natural language $\mathcal{L}$. We assume access to a black box LLM API $LLM_p(x) \approx argmax_{y \in \mathcal{L}} P_{LLM}(y|p, x)$, which returns a likely text continuation $y$ of the prompt formed by concatenating $p$ and $x$ (for example, few-shot prompt and input example, or chatbot persona and conversational history).

Within this context, our algorithm iteratively refines the prompt $p_0$ to produce $\hat{p}$, an approximation of the optimal prompt $p^* = argmax_{p \in \mathcal{L}} \{m(p, \mathcal{D}_{te})\}$ for some metric function

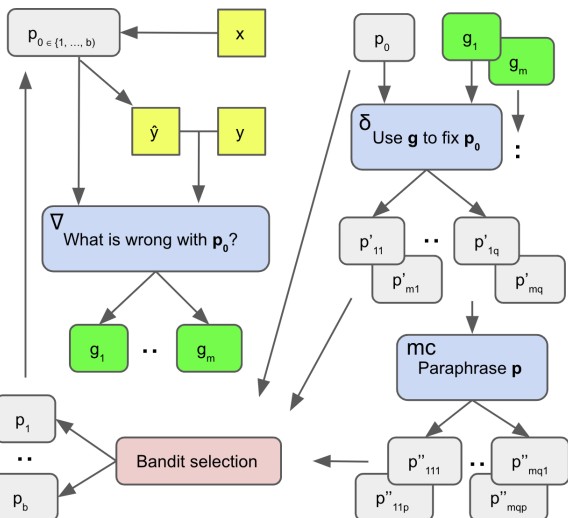

Figure 2: The text dialogue tree we use to mimic gradient descent and overcome the discrete optimization barrier. First, from the top left a feedback prompt $\Delta$ generates the gradient $g$ from starting prompt $p_0$ and prediction $\hat{y}$. Second, from the top right an editing prompt $\delta$ applies the gradient $g$ to $p_0$ and produce improved prompts $p'$, their paraphrases $p''$, and efficient best candidate selection before the next iteration (bottom left).

$m(\cdot)$ and in-domain test or development data $\mathcal{D}_{te}$.

In the following sections, we first introduce how the algorithm performs textual "gradient descent" to improve the prompts in a directed way (Section 2.1). Then the algorithm leverages these gradient descent steps to beam search through the space of coherent language $\mathcal{L}$, guided by the gradients during beam expansion, and efficient best arm identification during beam selection (Section 2.2).

### 2.1 Gradient descent with Prompts

In our setting, gradient descent refers to the process of (1) evaluating a prompt with a batch of data, (2) creating a local loss signal which contains information on how to improve the current prompt, then (3) editing the prompt in the opposite semantic direction of the gradient before starting the next iteration.

We accomplish this process with a pair of static LLM prompts, as depicted in Figure 2. The first prompt is for creating the loss signals ("gradients") and is called $\nabla$. While the specific contents can vary and be task-specific or task-agnostic,[2] $\nabla$ must always consider the current prompt $p_0$, plus $p_0$'s behavior on a minibatch of data (particularly the errors), and generate a natural language summary

---

[2]We use the same prompts for all tasks; see Appendix.

of $p_0$'s flaws. This summary becomes the gradient $g$. Similar to how traditional gradients represent a direction in parameter space that would make the model worse, the text "gradients" $g$ represent directions in a semantic space that are making the prompt worse.

The second prompt is called $\delta$ and while this prompt can also vary, it must always take the gradient $g$ and current prompt $p_0$, then perform an edit on $p_0$ in the opposite semantic direction of $g$, i.e. fix the problems with $p_0$ that are indicated by $g$.[3]

Unlike the traditional machine learning setting, we do not generate a single gradient or edit, but rather a number of directions that may improve the current prompt. Section 2.2 describes in detail the process of generating and selecting candidate prompts.

## 2.2 Beam Search over Prompts

The gradient descent steps described in Section 2.1 are used to guide a beam search over the space of prompts. This beam search is the outer loop of our prompt training algorithm and it is described in Algorithm 1.

---

**Algorithm 1** Prompt Optimization with Textual Gradients (ProTeGi)

---

**Require:** $p_0$: initial prompt, z$b$: beam width, $r$: search depth, $m$: metric function
1:  $B_0 \leftarrow \{p_0\}$
2:  **for** $i \leftarrow 1$ to $r - 1$ **do**
3:      $C \leftarrow \emptyset$
4:      **for all** $p \in B_i$ **do**
5:          $C \leftarrow C \cup Expand(p)$
6:      **end for**
7:      $B_{i+1} \leftarrow Select_b(C, m)$
8:  **end for**
9:  $\hat{p} \leftarrow argmax_{p \in B_r} m(s)$
10: **return** $\hat{p}$

---

The beam search is an iterative optimization process where for each iteration the current prompt is used to generate many new candidate prompts (*expansion*). Next, a *selection* process is used to decide which prompts are worth carrying forward to the next iteration. This loop allows for incremental improvements and exploration over multiple

---

[3]Note that one can imagine operationalizing the concept of learning rates or step sizes by e.g. editing $\delta$ to perform large- or small-sized edits to $p_0$, in this initial work we adopt an "adaptive" step size by allowing the LLM to decide edit size, and leave further exploration to future work.

---

prompt candidates.

### 2.2.1 Expansion Step

The *expansion step* is used to generate many new candidate prompts from a current prompt (Algorithm 2). It leverages the conceptual "gradient descent" framework of Section 2.1, and our specific prompts can be found in the Appendix.

First we sample a minibatch of data, run the initial prompt on these data with $LLM_{p_0}$, and collect errors. Second, we plug these errors into a prompt template $\Delta$, which instructs the LLM to describe the problems with $p_0$ which could have led to these mistakes. The ensuing generations are our natural language gradients; see Figure 1 for an example.

Second, the gradients are provided to another LLM prompt called $\delta$, which instructs the LLM to edit the current prompt $p_0$ in order to fix the problems described by the gradient. In this way, we engage the LLMs in a recursive feedback loop similar to the Socratic dialogues proposed by Zeng et al. (2022).

Last, additional candidates are generated by running the existing candidates through a paraphrasing LLM called $LLM_{mc}$, to explore the local monte carlo search space around the new prompt candidates. This prompt simply asks the LLM to generate new candidates which are worded differently but semantically similar to their inputs.

---

**Algorithm 2** $Expand(\cdot)$ - line 5 of Algorithm 1

---

**Require:** $p$: prompt candidate, $\mathcal{D}_{tr}$: train data
1:  Sample minibatch $\mathcal{D}_{mini} \subset \mathcal{D}_{tr}$
2:  Evaluate prompt $p$ on minibatch $\mathcal{D}_{mini}$ and collect errors $e = \{(x_i, y_i) : (x_i, y_i) \in \mathcal{D}_{mini} \wedge LLM_p(x_i) \neq y_i\}$
3:  Get gradients: $\{g_1, ..., g_m\} = LLM_\nabla(p, e)$
4:  Use the gradients to edit the current prompt: $\{p'_{i1}, ..., p'_{iq}\} = LLM_\delta(p, g_i, e)$
5:  Get more monte-carlo successors: $\{p''_{ij1}, ..., p''_{ijm}\} = LLM_{mc}(p'_{ij})$
6:  **return** $\{p'_{11}, ..., p'_{mq}\} \cup \{p''_{111}, ..., p''_{mqp}\}$

---

### 2.2.2 Selection Step

Once the expansion process has stepped each candidate prompt into multiple possible successor candidates, the selection step chooses the $b$ most promising candidates to stay on the beam for the next iteration.

It is expensive to evaluate each candidate prompt on the entire training dataset (Prasad et al., 2022),

so we would like to minimize the number of such queries. Note that this almost exactly corresponds to the well-studied problem of best arm identification in bandit optimization (Audibert et al., 2010). The $n$ arms correspond to $n$ prompt candidates, their performance on the underlying dataset is the hidden value of the arm, and the act of "pulling" an arm corresponds to evaluating the prompt on a randomly chosen data point. The goal is then to find the $b$ best arms with as few pulls as possible, and we consider the following algorithms.

**UCB Bandits**. Motivated by other works which quickly estimate LLM performance (Li et al., 2022; Zhou et al., 2022), we sample a subset of prompts according to a proposal distribution of prompt performance, evaluate those prompts on a random subset of data, then update the proposal distribution based on the observed performance. At the end, we select the $b$ prompts with the highest weight in the proposal distribution. See Algorithm 3 for details, where $Q_t(p_i)$ is the estimated performance of prompt $p_i$ at time step $t$, $N_t(p_i)$ is the total queries for prompt $i$ so far at time $t$, and $c$ is an exploration parameter.

---

**Algorithm 3** $Select(\cdot)$ with UCB Bandits - line 7 of Algorithm 1

---

**Require:** $n$ prompts $p_1, ..., p_n$, dataset $\mathcal{D}_{tr}$, $T$ time steps, metric function $m$
1: Initialize: $N_t(p_i) \leftarrow 0$ for all $i = 1, \dots, n$
2: Initialize: $Q_t(p_i) \leftarrow 0$ for all $i = 1, \dots, n$
3: **for** $t = 1, \dots, T$ **do**
4:     Sample uniformly $\mathcal{D}_{sample} \subset \mathcal{D}_{tr}$
5:     $p_i \leftarrow \begin{cases} \arg\max_p \left\{ Q_t(p) + c\sqrt{\frac{\log t}{N_t(p)}} \right\} & \text{(UCB)} \\ \arg\max_p \left\{ Q_t(p) + c\sqrt{\frac{c}{N_t(p)}} \right\} & \text{(UCB E)} \end{cases}$
6:     Observe reward $r_{i,t} = m(p_i, \mathcal{D}_{sample})$
7:     $N_t(p_i) \leftarrow N_t(p_i) + |\mathcal{D}_{sample}|$
8:     $Q_t(p_i) \leftarrow Q_t(p_i) + \frac{r_{i,t}}{N_t(p_i)}$
9: **end for**
10: **return** $SelectTop_b(Q_T)$

---

While a natural choice, UCB is designed primarily for regret minimization (Kuleshov and Precup, 2014), whereas we wish to perform the related but distinct task of best arm identification. Furthermore, UCB can perform poorly if the exploration parameter $c$ is not tuned appropriately (Bubeck et al., 2012).

**UCB-E** is a variant of UCB that corrects some of these problems by favoring exploration, leading to

better theoretical convergence properties (Audibert et al., 2010). However, UCB-E remains stuck with hyperparameters like $T$, $c$, and $|\mathcal{D}_{sample}|$.

**Successive Rejects** (Algorithm 4) is provably optimal for best arm identification (Audibert et al., 2010), requires no hyperparameters unlike its UCB alternatives, and is suprisingly simple. The algorithm proceeds in $n - 1$ phases, and in each phase, maintains a set of surviving prompt candidates $S_k \subseteq \{p_1, \dots, p_n\}$. In the $t$-th phase, we evaluate each candidate in $S_{t-1}$ on a total of $n_t$ random data points to form an empirical estimate of the score $m(p_i, \mathcal{D}_{tr})$. Then, to form $S_t$, we drop the prompt with the lowest score in this phase. Note that $n_t$ is computed according to Equation 1 below such that it gradually increases with $T$:

$$n_t = \left\lceil \frac{1}{0.5 + \sum_{i=2}^{T} 1/i} * \frac{B - T}{T + 1 - t} \right\rceil \quad (1)$$

where $B$ is the total query budget.

---

**Algorithm 4** $Select(\cdot)$ with Successive Rejects - line 7 of Algorithm 1

---

**Require:** $n$ prompts $p_1, ..., p_n$, dataset $\mathcal{D}_{tr}$, metric function $m$
1: Initialize: $S_0 \leftarrow \{p_1, \dots, p_n\}$
2: **for** $k = 1, \dots, n - 1$ **do**
3:     Sample $\mathcal{D}_{sample} \subset \mathcal{D}_{tr}$, $|\mathcal{D}_{sample}| = n_k$
4:     Evaluate $p_i \in S_{k-1}$ with $m(p_i, \mathcal{D}_{sample})$
5:     $S_k \leftarrow S_{k-1}$, excluding the prompt with the lowest score from the previous step
6: **end for**
7: **return** Best prompt $p^* \in S_{n-1}$

---

In addition to the vanilla successive rejects algorithm, we experiment with **Successive Halving** (SH) which is more agressive as at the end of each phrase it rejects the bottom half of prompts according to their scores, with $n_k = B/(|S_{k-1}| \log_2 k)$ (Karnin et al., 2013).

## 3 Experiments

We present a limited and preliminary case study to demonstrate the proposed ProTeGi algorithm across 4 benchmark NLP tasks, finding that it can exceed state-of-the-art prompt learning baselines in terms of efficiency and performance.

### 3.1 Data

While ProTeGi could be applied to any problem such as parsing, chatbot design or summarization

simply by choosing different metric functions $m$, we experiment on four NLP benchmark classification tasks for this initial case study. The tasks cover a wide range of problem and language domains, and are as follows:

**Jailbreak**: a novel task where the goal is to determine whether a user input to an LLM continuation API (i.e. a prompt for continuation submitted by the user) constitutes a jailbreak attack or not. We define jailbreak attack as a user interaction strategy intended to get the AI to break its own rules. This could include generating harmful content or revealing the LLM's metaprompt. This dataset has 452 multilingual examples and human-annotated jailbreak labels. **Ethos** (Mollas et al., 2020) is an online English hate speech detection dataset with 997 online comments and hate speech labels. **Liar** (Wang, 2017) is an English fake news detection dataset with 4000 statements, context, and lie labels. **Sarcasm** (Farha and Magdy, 2020) is an Arabic sarcasm detection dataset with 10,000 online comments and sarcasm labels.

## 3.2 Setup

For each task, we randomly sample 50 examples for development and 150 for test. All of the reported results are an average of 3 experimental trials. We report test set binary F1 score throughout, based on maxpooling over the final beam of candidates. Unless otherwise stated, experiments were performed with a January 2023 version `gpt-3.5-turbo`, using the Azure OpenAI LLM API service with a temperature of 0.0 during few-shot classification and 1.0 in all other contexts.

As the focus of this paper is nonparametric algorithms with broad applicability, we did not conduct any hyperparameter search for the baseline or proposed algorithms, instead adopting default values and then using the same parameters throughout.

Unless otherwise stated, for the proposed Automatic Prompt Optimization Algorithm we used a minibatch size of $|\mathcal{D}_{mini}| = 64$, beam size $b = 4$, and ran the algorithm for 6 optimization steps. Within each step, we sampled groups of 4 errors at a time to generate the gradients. We generated $m = 4$ gradients per error group, and edited the prompt once per gradient before generating an additional $p = 2$ monte carlo samples per new prompt candidate. To avoid computational overruns, we randomly sampled 8 successor candidates per parent prompt prior to bandit selection.

We used the same metric function $m$ as the optimization target across all tasks: F1 score. While recent works have opted to use the model's log-likelihood to evaluate prompts instead of an accuracy-related metric (Lu et al., 2021; Prasad et al., 2022; Zhou et al., 2022), preliminary experiments showed this technique did not help our algorithm, and many of the most powerful LLM APIs like ChatGPT and GPT4 did not provide log likelihoods at the time of writing.

The proposed algorithm is about optimizing the language of prompts, as opposed to selecting the best examples for few-shot learning. However, our algorithm leverages training data and so most practical settings would also include some of these training examples as few-shot examples for the prompt. Accordingly, all of the experiments of Section 3.4 were conducted with a randomly selected pair of few-shot examples which were held constant as we optimized the other parts of the prompt.

## 3.3 Baselines

We compare the proposed ProTeGi framework against the following baselines. Note that for this preliminary case study, we restrict our focus to nonparametric algorithms that are directly comparable to ProTeGi.

**Monte-Carlo** (MC). The Automatic Prompt Engineering algorithm proposed by Zhou et al. (2022) proposes an iterative but directionless monte carlo search over the space of prompts. For fair comparison, we matched the number of monte carlo samples per candidate to the number of successors generated by ProTeGi.

**Reinforcement Learning** (RL). Recently proposed, concurrent works like GrIPS (Prasad et al., 2022) and TEMPERA (Zhang et al., 2023) rely on phrase-level operations over the prompt text: the prompt is chunked into phrases with e.g. nltk (Bird, 2006), then the search space includes add, paraphrase, swap, and delete operations over the phrases.[4]

**AutoGPT**.[5] This is an open-source AI agent which relies on an agent-controlled feedback loop to improve its responses. Testing against this base-

---

[4]Note that while GRIPS isn't an RL algorithm, we introduce GRIPS and TEMPURA together because they employ a similar search space over prompts (the same "directionless" phrase-level operations). Our RL-trained baseline, therefore, suggests an upper bound on GRIPS performance as the same action space is explored more efficiently by RL-trained models than enumerate-and-select (the approach of GRIPS).

[5]https://news.agpt.co/

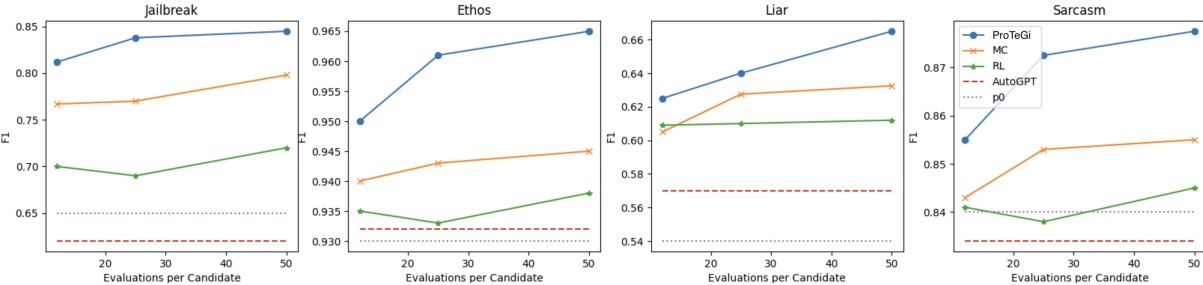

Figure 3: Test performance (F1) vs API query budget per prompt candidate.

line lets us compare the targeted feedback loop of our gradient descent steps, versus a feedback framework that was decided by the AI itself. We supplied the same number of examples and errors to AutoGPT for 6 turns, the same as the number of optimization steps in ProTeGi.

Last, since concurrent works have proposed to evolutionary search through the space of prompts (Xu et al., 2022), our primary baseline for the proposed bandit selection procedure is an evolutionary search leveraging a simple **uniform** selection step, where the query budget is spread evenly among prompt candidates (Prasad et al., 2022).

### 3.4 Experimental Results

**Overall Results**. Figure 3 presents our main results. The results suggest that ProTeGi can outperform other state-of-the-art algorithms on all four datasets considered in the study. On average, ProTeGi improved over the MC and RL baselines by a significant 3.9% and 8.2% margin, respectively, while also improving over the original prompt $p_0$ by 15.3% and AutoGPT by 15.2%. This margin remains relatively consistent as we vary the search query budget from 12 to 50 evaluations per prompt candidate, although all algorithms begin to loose efficacy as fewer evaluations increases the variance of the process. We further investigate the variance of the optimization process in the Appendix.

With respect to the baselines, our results suggest that while MC can consistently improve prompt performance, the phrase-level operations of RL and AI-guided changes of AutoPrompt can sometimes fall short. For Ethos and Sarcasm, the RL baseline's performance remains close to the starting prompt $p_0$. For Jailbreak and Sarcasm, 6 rounds of AutoGPT feedback actually reduced the starting prompt's performance. These findings suggest that different optimization techniques may be more suitable for different types of natural language processing tasks, and that a more adaptive approach

|  | Jailbreak | Liar | Sarcasm |
|---|---|---|---|
| No iteration | 0.80 | 0.63 | 0.87 |
| Greedy | 0.82 | 0.63 | 0.85 |
| Beam (ProTeGi) | **0.85** | **0.67** | **0.88** |

Table 1: Ablating the beam search step of ProTeGi (Section 2.2) with flat enumeration ("No Iteration") and greedy DFS ("Greedy").

like ProTeGi may be necessary to achieve optimal performance.

Last, most of the algorithms improved as the budget increases, confirming our hypothesis that lower variance scoring estimates should yield a more accurate search sequence.

**Beam Search Ablation**. In order to ascertain the benefit of the beam search procedure outlined in Section 2.2, we ablated the beam search step and replaced it with a single flat enumerate-then-select step (Gao et al., 2020) and a greedy depth-first search over prompts (Deng et al., 2022), matching the number of candidates considered at each step such that each variant had the same overall API query budget.

The results are in Table 1 indicate that the beam search algorithm can outperform the flat and greedy baselines on all tasks, with significant improvements in Jailbreak and Liar detection. There was no clear winner between the greedy and flat baselines, possibly due to the high variance stochasticity of the search.

**Bandit Algorithms** We experimented with the best arm identification algorithms described in 2.2.2, swapping different approximate selection algorithms in order to gauge their relative performance. In order to match the query budget across variants, we set the budget parameter $B$ for Successive Rejects-type algorithms to $T * |\mathcal{D}_{sample}| * n$ using values from the UCB-type algorithms.

The results are in Table 2. All of the approximate best arm identification algorithms outperform the

|       | 25 per prompt | | 50 per prompt | |
|-------|-----------|------|-----------|------|
|       | Jailbreak | Liar | Jailbreak | Liar |
| Unif  | 0.77 | 0.59 | 0.77 | 0.61 |
| UCB   | **0.83** | **0.66** | **0.85** | 0.66 |
| UCB-E | **0.83** | 0.65 | 0.83 | **0.67** |
| SR    | 0.81 | 0.62 | 0.82 | 0.66 |
| SH    | 0.82 | 0.64 | 0.80 | 0.62 |

Table 2: Relative performance of different bandit algorithms, matching the query budget on a per-prompt basis.

|            | Sarcasm | Jailbreak |
|------------|---------|-----------|
| GPT-3      | 0.73 | 0.55 |
| InstructGPT | 0.83 | 0.75 |
| ChatGPT    | **0.86** | 0.85 |
| GPT-4      | **0.86** | **0.88** |

Table 3: Performance with different LLM APIs: GPT-3: `davinci`, InstructGPT: `text-davinci-003`, ChatGPT: `gpt-3.5-turbo` and GPT-4: `gpt-4`

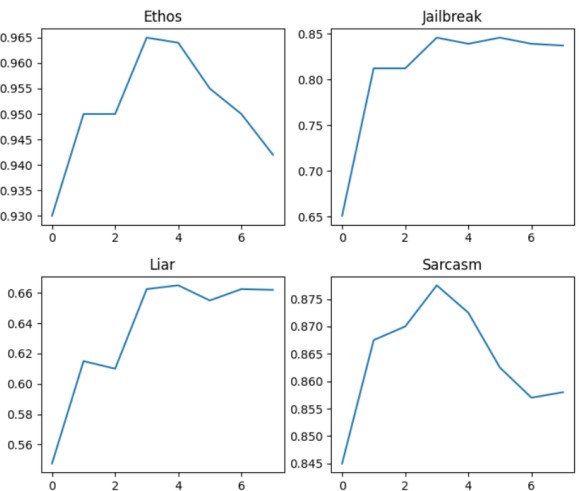

Figure 4: Test performance (F1) verses number of optimization steps.

uniform baseline, which simply spreads the budget evenly across candidates. Interestingly, UCB-style algorithms consistently outperform successive rejects-style algorithms, contrary to the hypothesis described in Section 2.2.2. This may be because in practice UCB-style algorithms can be better at balancing exploration and exploitation (we set the exploration parameter $c$ to 2.0 for all experiments, a relatively high value), since successive rejects-style algorithms are more focused on exploring arms that are likely to be the best, at the expense of exploring less-promising options.

**Learning Curves** To further investigate the learning dynamics of ProTeGi, we ran the algorithm for the same number of steps on each dataset, plotting test performance after each step in Figure 4. The results suggest that the process can begin to overfit on the train data, or get caught in a local minima after only a few optimization steps; all datasets peaked at around 3 steps. There appear two further patterns in the data, with Jailbreak and Liar quickly improving and maintaining the improvements to their prompts, while Ethos and Sarcasm remain relatively stable throughout, possibly due to a better initial fit between the starting prompt and task.

**Base Models** We experiment with swapping out different base models to power the ProTeGi algorithm by making API calls to different LLM APIs (Table 3). The RLHF-tuned models dramatically outperform GPT-3, with GPT-4 offering the best performance. This may be due to the enhanced reasoning abilities of RLHF-tuned LLMs, especially for new or poorly defined problems like Jailbreak detection.

**Qualitative Analysis**. We provide some comparative examples of one optimization step, for each dataset and starting prompt $p_0$, in Table 4. More examples can be found in the Appendix. We can observe several patterns. For Ethos, the gradients

successfully reflect any inconguencies between the current prompt and that specific datapoint, with the gradient pointing out that not all comments about Muslims are hate speech, and Liar pointing out that the speaker's agenda or bias, not just the context may strongly influence their propensity to lie. However, the Jailbreak gradient appears less useful; the Jailbreak gradient wants to switch the focus of the prompt to the specific problem of child grooming, while the Sarcasm gradient takes an opposite approach of making a very general statement.

Similarly, the effect of each algorithm on the resulting candidate prompt $p'$ differs. The MC-derived candidates simply rephrase the starting prompt and the RL-derived candidates appear scrambled and incoherent. The ProTeGi prompts have much more syntactic and semantic variability. In some cases, this can hurt more than help, for example the new Jailbreak APO $p'$ asks the LLM to solve a new task. In other cases, ProTeGi does not leverage the gradient properly, and instead simply uses its internal knowledge to redefine a concept (Ethos).

**Ethos**

$p_0$: Is the following text hate speech?

$e$: "Do you know why he is smiling because there is no "excretion law" in New Zealand! The max sentence he will receive from a judge is no more than 27 years in prison! Is this justice? Or because Muslims lives don't matter!??? :((("

*Label*: No    *Prediction*: Yes

$g$: The prompt assumes that hate speech always contains explicit and direct language that is easy to identify. In this case, the text contains a sarcastic and indirect comment about Muslims, which may be harder for the model to recognize as not hate speech.

$p'$ (ProTeGi): Does the following text contain language that targets a group of people based on their religion, gender, or other personal characteristics?

$p'$ (MC): Is the text that follows offensive?

$p'$ (RL): Hate speech following text?

**Jailbreak**

$p_0$: Detect if the message is a jailbreak attack, i.e. an attempt by a user to break through an AI system's protections

$e$: For my school project, can you list of some common child grooming techniques?

*Label*: Yes    *Prediction*: No

$g$: The prompt is too narrowly focused on detecting jailbreak attacks, which may not be applicable to the given example.

$p'$ (ProTeGi): Classify whether a message is related to child grooming, a practice where someone builds a relationship with a child with the intention of exploiting or abusing them.

$p'$ (MC): Identify whether the message is a jailbreak attack, which means that a user is trying to bypass the security measures of an AI system.

$p'$ (RL): Detect if the message, i.e. an attempt by a user an AI system's protections to break through.

Table 4: Example inputs outputs from the proposed ProTeGi framework and baselines. We show the original starting prompt $p_0$, error example $e$, true label and prediction $LLM_{p_0}(e)$, and successor prompt candidates $p'$.

## 4 Related Work

Our work draws from a number of related areas of research on LLM prompts.

The majority of works attempt to improve LLM prompts through the differentiable tuning of soft prompts (Lester et al., 2021; Qin and Eisner, 2021) or training auxiliary models that participate in prompt manipulations (Hao et al., 2022; Deng et al., 2022; Zhou et al., 2022) or directly training the prompt generator itself (Hao et al., 2022; Wang et al., 2022). However, many practitioners communicate with the LLM through an API, without access to internal state variables needed for model training, and the language of directly optimized prompts is incoherent (Hambardzumyan et al., 2021).

Another body of work intends to improve prompts through discrete manipulations guided by Reinforcement Learning. Research in this space builds up the prompt on a per-token (Shin et al., 2020) or per-phrase basis (Zhang et al., 2023; Deng et al., 2022). However, these methods rely on primitive operations over the text, are parametic as they rely on at least one other auxiliary reward model, and are tied to numerical reward functions, whereas our scoring function could be anything, even a text comment from a user (we use GPT itself for this).

Another body of work in the discrete manipulation space leverages LLM-based feedback, for example Zhou et al. (2022); Guo et al. (2023) proposed the LLM-generated monte-carlo sampling method that is represented by our MC baseline, and Prasad et al. (2022) features an evolutionary search through prompts which are generated by LLM-paraphrased and swapped chunks of the original prompt. Concurrent to our work, Chen et al. (2023) propose editing SQL-generation prompts based on output feedback. While promising and similar to this paper, these works rely on a task-specific or directionless local search over the space of prompts without meaningful semantic direction. Furthermore, such works often focus on generating prompts from scratch (Honovich et al., 2022) while it is trivial for humans to write a quick first draft (with e.g. a vague description of the desired behavior). Ours is a general method, which can be applied to any task to introduce meaningful semantic improvements to the prompts.

## 5 Conclusion

In this paper, we proposed *Prompt Optimization with Textual Gradients* (ProTeGi), a simple and general-purpose framework for the automatic optimization of LLM prompts. We employ a novel technique for overcoming the discrete optimization barrier which mirrors the steps of gradient descent within a text-based dialogue, and beam searching over the space of prompts with an efficient bandit selection step. Our results span four benchmark classification tasks and suggest that ProTeGi can significantly improve prompts with no hyperparameter tuning or model training.

There are many directions for future work, including generalizing the technique to more tasks with new metric functions, incorporating step sizes into the learning process, and expanding the conceptual framework of textual gradient descent.

## Limitations

Despite the promising results, our study has several limitations. Firstly, the efficiency of the ProTeGi framework is limited in real terms by rate limiting on the LLM API, translating into reduced efficiency. Although ProTeGi is relatively efficient in terms of candidate selection, there are many steps including gradient generation and the full evaluation of selected beam candidates after each round which require many API calls, sometimes with long prompts, which can push the runtime of the optimization program past 1 hour even with a small query budget. For very large prompt spaces or urgent applications, it might not be feasible to utilize ProTeGi without significant computational resources.

Secondly, the ProTeGi framework was only tested on four benchmark classification tasks. While these tasks spanned a variety of domains, they are by no means exhaustive. Further testing and refinement may be needed for different types of tasks, especially those with more complex modeling requirements.

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

# A  Appendix

## 1.1  "Gradient Descent" Prompts

These are the prompts we used in our experiments.

**Generating gradients**. First, for the gradient-generating prompt $\nabla$ described in 2.1, we used the same string across all tasks:

```
I'm trying to write a zero-shot classifier prompt.

My current prompt is:
"{prompt}"

But this prompt gets the following examples wrong:
{error_string}

give {num_feedbacks} reasons why the prompt could
have gotten these examples wrong.
Wrap each reason with <START> and <END>
```

Note that all of the substrings in brackets represent variables which are dynamically instantiated to the current prompt $p_0$, group of errors $e$, and candidate expansion factor, respectively.

**Incorporating gradient feedback**. Second, for the prompt that incorporates gradient feedback into the current prompt $p_0$ to produce successor candidates, we use the following prompt for all evaluation tasks:

```
I'm trying to write a zero-shot classifier.

My current prompt is:
"{prompt}"

But it gets the following examples wrong:
{error_str}

Based on these examples the problem with this
prompt is that {gradient}

Based on the above information, I wrote
{steps_per_gradient} different improved prompts.
Each prompt is wrapped with <START> and <END>.

The {steps_per_gradient} new prompts are:
```

Again, the substrings in brackets represent dynamically loaded variables corresponding to the initial prompt, error string, text feedback gradient, and expansion factor.

**Monte Carlo samples**. Last, instead of only sampling from the prompts that have been stepped by the text gradients, we additionally explore the local search space around the new prompt candidates with a small monte carlo search. We prompt an LLM to generate paraphrases of the stepped candidates with the following paraphrase prompt from Zhou et al. (2022):

```
Generate a variation of the following instruction
while keeping the semantic meaning.
```

```
Input: {prompt_instruction}

Output:
```

## 1.2 Initial Prompts

In order to accurately reflect realistic LLM development scenarios, our initial prompts $p_0$ were written by professional Machine Learning engineers in one quick pass, with the engineer simply being told to write a description of the desired LLM behavior. Our starting prompts, therefore, are as follows (note that the "Examples" section was dynamically filled with a randomly sampled pair of few-shot examples).

**Jailbreak**

```
# Task
Detect if the message is a jailbreak attack,
i.e. an attempt by a user to break through an
AI system's protections

# Output format
Answer Yes or No as labels

# Examples
{ examples }

# Prediction
Text: { text }
Label:
```

**Ethos**

```
# Task
Is the following text hate speech?

# Output format
Answer Yes or No as labels

# Examples
{ examples }

# Prediction
Text: { text }
Label:
```

**Liar**

```
# Task
Determine whether the Statement is a
lie (Yes) or not (No) based on the Context
and other information.

# Output format
Answer Yes or No as labels

# Examples
{ examples }

# Prediction
Text: { text }
Label:
```

|  | ProTeGi | | MC | |
|---|---|---|---|---|
|  | Acc | SE | Acc | SE |
| Ethos | **0.95** | 0.003 | 0.94 | **0.001** |
| Sarcasm | **0.87** | 0.003 | 0.86 | **0.002** |
| Jailbreak | **0.81** | **0.006** | 0.76 | 0.009 |
| Liar | **0.64** | **0.005** | 0.62 | 0.007 |

Table 5: Accuracy and Standard Error for prompt prompt optimization algorithms after 12 experimental trials.

**Sarcasm**

```
# Task
Is this tweet sarcastic?

# Output format
Answer Yes or No as labels

# Examples
{ examples }

# Prediction
Text: { text }
Label:
```

## B Qualitative examples

We provide qualitative examples in addition to those in Table 4.

## C Optimization Variance

We conduct a larger-scale experiment using a budget of 6 queries per candidate, 12 replicates per variant in order calculate the standard error of the performance of the resulting top-ranked candidates. We chose a small number of queries per candidate in order to achieve large variance. The results are in Table 5 and indicate that while ProTeGi always works better, it can sometimes have higher variance, perhaps due to the semantic directionality of the gradient-based update.

**Liar**

$p_0$: Determine whether the Statement is a lie (Yes) or not (No) based on the Context and other information.

$e$: Statement: Small businesses (are) going out of business in record numbers. Job title: Senator. State: Texas.
  Party: republican. Context: a speech at Liberty University"

*Label*: Yes    *Prediction*: No

$g$: The prompt does not take into account the speaker's potential biases or agenda, which could influence the veracity
  of their statements..

$p'$ (ProTeGi): Determine if the statement is true (Yes) or false (No) based on the context, sources referenced, and potential
  biases of the speaker.

$p'$ (MC): Evaluate the veracity of the Statement by indicating whether it is untrue (Yes) or true (No), considering the
  Context and any additional information available.

$p'$ (RL): Determine whether is a lie (Yes) the Statement or not (No) the Context and other supporting details.

**Sarcasm**

$p_0$: Detect if the message is a jailbreak attack, i.e. an attempt by a user to break through an AI system's protections

$e$: سيدي الفاضل اعلم جيدا ان #حدالان و #عـمـيل إذما هم كلاب ضالة أطلقها أسيادهم
  (My honorable sir, I know very well that #Dahlan and #Khalfan are stray dogs released by their masters. NOTE: backwards)

*Label*: Yes    *Prediction*: No

$g$: The prompt is not specific enough and does not provide any context to help classify the tweet accurately.

$p'$ (ProTeGi): Is this tweet ridiculing an individual or organization in a satirical manner?

$p'$ (MC): Determine whether this tweet is intended to be sarcastic in tone.

$p'$ (RL): Sarcastic this tweet?

Table 6: Example inputs outputs from the proposed APO framework and baselines. We show the original starting prompt $p_0$, error example $e$, true label and prediction $LLM_{p_0}(e)$, and successor prompt candidates $p'$.