# OpenReview forum: "Automatic Prompt Optimization with "Gradient Descent" and Beam Search"
_EMNLP/2023/Conference — EMNLP 2023 Main_

### Official Review · Reviewer_cYMM · 2023-08-04

**Soundness:** 4

**Excitement:**

4: Strong: This paper deepens the understanding of some phenomenon or lowers the barriers to an existing research direction.

**Paper Topic And Main Contributions:**

This paper presents a framework, Prompt Optimization with Textual Gradients (ProTeGi), for automatic LLM prompt optimization. Starting from an initial prompt, it iteratively applies LLM with predefined meta-prompt over randomly sampled minibatch of examples to generate feedbacks as gradients. It then uses another meta-prompt with the initial prompt and gradients to produce improved prompts. Finally, it uses Bandit Selection algorithms to prune the size of candidate prompts (beam search). Empirical studies show the proposed method outperforms baselines like Monte Carlo search, reinforcement learning, and AutoGPT. Further studies also revealed the impact from beam search and bandit algorithms.

**Questions For The Authors:**

1. Table 1: it would also be good to show how beam size impact the performance.
2. Line 322, although it states the log-likelihood does not help the proposed method, it is still be interesting to show how the log-likelihood change with iterations.

**Reasons To Accept:**

1. The task discussed in this paper has high impact in both academic and industry.
2. The paper presents a framework that is generally applicable to many LLM based methods. The analyses also shows interesting properties of the proposed method, which might lead to more followup studies.
3. The paper is well-organized and easy to read.

**Reasons To Reject:**

Some important details are not clearly revealed in the paper. For example, it would be interesting to see what the feedback prompt and editing prompt are, and how they impact the whole framework.

**Reproducibility:**

3: Could reproduce the results with some difficulty. The settings of parameters are underspecified or subjectively determined; the training/evaluation data are not widely available.

**Reviewer Confidence:**

2: Willing to defend my evaluation, but it is fairly likely that I missed some details, didn't understand some central points, or can't be sure about the novelty of the work.

---

> ### Author Rebuttal · Authors · 2023-08-28
>
> Thank you for taking the time to review our paper, we agree that this is a novel and powerful solution to an important problem.
>
> Our feedback and editing prompts can be found in the Appendix A (sup materials). We agree that studying the effect of these prompts (or even some meta-optimization on these prompts) is important. For this initial work we simply fixed these prompts in order to demonstrate that the novel technique (text optimization with LLM-written "gradients") is feasible. We will include results on how these prompts affect the whole framework in the camera ready.
>
> We will certainly include results on beam size and log likelihood for the camera ready.

---

### Official Review · Reviewer_jdT4 · 2023-08-04

**Soundness:** 3

**Excitement:**

4: Strong: This paper deepens the understanding of some phenomenon or lowers the barriers to an existing research direction.

**Paper Topic And Main Contributions:**

- This paper proposes a new framework to generate, then edit, and then paraphrase and score new candidate prompts for large language models. It first generates reasoning for the wrong labeling for the model errors. This reasoning has been interpreted as textual gradients. Then it receives these reasons to perform edits over the initial prompt. Later, the candidate prompts are paraphrased to include more diverse alternatives. Throughout the search, the method keeps a beam of candidate prompts and evaluates them efficiently for the next beam iteration. It successfully uses USB bandits to evaluate the candidate prompts within a beam over a training dataset efficiently.

**Questions For The Authors:**

- Is it OK to interpret your textual gradients as kind of chain-of-thoughts to generate new prompts using the $LLM_{\delta}(p, g_i, e)$?

- Can the authors expand the details of hyper-parameters for some of the baselines? The following paragraph (Line 300) in your paper seems a bit concerning regarding the experimental results:
"As the focus of this paper is non-parametric algorithms with broad applicability, we did not conduct any hyper-parameter search for the baseline or proposed algorithms, instead adopting default values and then using the same parameters throughout."
You are selecting 150 examples as your internal validation. Why is it not possible to tune the hyper-parameters of the baselines? What is the "compelling" reason to stay fully non-parametric while tuning few hyper-parameters might impact the results significantly?

- What would be the performance of your models using just $LLM_{mc}$ without generating textual gradients + edits and just purely relying on another prompted LLM to rephrase the initial $P_0$?

**Reasons To Accept:**

- The paper has a novel interpretation as the "textual gradient" and "moving in the opposite direction of the textual gradient" to make local edits in the semantic space. This has been explained well enough and the method clearly outperforms previous baselines on their 4 datasets.

- As shown by the examples in the paper (Table 4), the generated prompts by this method are more understandable and contains more information about the expected context/output and outperforms the optimized prompts given another baselines.

**Reasons To Reject:**

- The proposed method to find better performing prompts is applicable only on truly large language models as they are capable of generating the textual gradients or reasoning steps for wrong generation by just prompting the LLM. The technique might not be useful for medium-size LMs such as T5-large or BERT-large, however previous discrete gradient-free prompt optimization techniques such as GrIPs or RLPrompt can be applied generally to any LM and they have been tested with medium-size LMs. It would be great to see how your concept of textual gradients can be extracted or applied from/to medium-size LMs.

**Reproducibility:**

4: Could mostly reproduce the results, but there may be some variation because of sample variance or minor variations in their interpretation of the protocol or method.

**Reviewer Confidence:**

4: Quite sure. I tried to check the important points carefully. It's unlikely, though conceivable, that I missed something that should affect my ratings.

**Typos Grammar Style And Presentation Improvements:**

- Fix typo in Algorithm 1 $z$b, what is z?

- The mention of GrIPS within an RL technique for prompt optimization is not correct! The technique is purely an edit based method and then selection of new candidate prompts on a search set based on the final task performance + entropy of predictions. Regarding the phrase chunking of GrIPS, it does not use pure NLTK! They use phrase chunking based on another CRF-based constituency parser. Please modify the description of the method correctly.

---

> ### Author Rebuttal · Authors · 2023-08-28
>
> Thank you for taking the time to review our paper and for your insightful comments. We appreciate your positive remarks on the novel interpretation of "textual gradients" and how our method outperforms the baselines across the datasets.
>
> Reasons to Reject:
>
> Applicability to Medium-Size Language Models: You raised a valid concern. While the paper focused on large language models like GPT3, chatGPT and GPT4, we agree that exploring the method's applicability to medium-sized models would be valuable. We tried it with BART-large (25 evaluations per candidate) and noticed that the performance breaks down significantly, possibly due to poor quality gradients and edits. We will include a more detailed study in the camera ready and in future work:
> https://imgur.com/t7c7YaD
>
>
> Questions:
>
> Interpretation of Textual Gradients: Yes, it is accurate to view textual gradients as a chain-of-thought mechanism to generate new prompts. They serve as a reasoning step for making local semantic adjustments.
>
> Hyper-parameters for Baselines: We appreciate the concern regarding hyper-parameters. The decision to use default values for the baselines was in line with our focus on non-parametric algorithms for broader applicability. Most practitioners seek a "plug and play" solution which can improve their prompts without any tuning. However, we acknowledge that hyper-parameter tuning might significantly impact results and plan to include a discussion on this in the revised manuscript.
>
> Performance Without Textual Gradients: using LLM_mc alone to rephrase p_0 would be similar to the MC baseline (see section 3.3: https://arxiv.org/abs/2211.01910).
>
> Typographical and Stylistic Concerns:
>
> Algorithm 1:  "zb" is a typo, we meant to just write "b" for beam width. Thank you.
>
> GriPS: Thank you! We do apologize for this confusion and will update our description accordingly. We introduced them together because GRIPS and TEMPURA employ a similar search space over prompts (the same phrase-level operations: delete, swap, paraphrase, add), and so GRIPS has similar fundamental characteristics as TEMPURA (do you want to choose from a discrete pool of edit operations using an RL-trained orchestrator (TEMPURA), or performance + prediction entropy (GRIPS) ). We will certainly correct this for the camera ready.
>
>
> We are grateful for your constructive feedback, which will greatly help us improve the quality of our paper!

---

### Official Review · Reviewer_2zD3 · 2023-08-05

**Typos Grammar Style And Presentation Improvements:** 1. In Figure 2, where is $\Delta$ men…
**Soundness:** 4

**Excitement:**

3: Ambivalent: It has merits (e.g., it reports state-of-the-art results, the idea is nice), but there are key weaknesses (e.g., it describes incremental work), and it can significantly benefit from another round of revision. However, I won't object to accepting it if my co-reviewers champion it.

**Paper Topic And Main Contributions:**

This paper proposes an automatic prompt optimization method by iteratively prompting an LM to improve upon previous prompts.
Using a set of labeled examples, the method starts with an initial prompt and iteratively query the language model for new prompts when errors occur. An upper confidence bound (UCB)-based algorithm is proposed to select new prompts.
The authors evaluate their method on 4 benchmarks using GPT3.5 and show the efficacy of their method.

**Questions For The Authors:**

* Line 351, GRIPS (Prasad et al. 2022) does not use reinforcement learning. Thus it should not be posited in that paragraph.

A. How does the proposed method compare to GRIPS in terms of performance?

B. In Figure 4, the performance on Ethos and Sarcasm drops after ~4 optimization steps. What is the reason for this?

**Reasons To Accept:**

The proposed method provides a flexible way to optimize prompts in a gradient-free manner using discrete feedback. It's effective and potentially useful for many other tasks that require manual prompt engineering.

**Reasons To Reject:**

Gradient-free iterative prompt search has been explored in previous work (e.g., RLPrompt, GRIPS, TEMPERA). It is important to emphasize the novelty and contributions of this work and provide both qualitative and quantitative comparisons with previous work. However, such comparisons are missing from the paper. I.e., what kind of errors can be fixed by the proposed method but not by previous work? What are some example optimization sequences of GRIPS and the proposed method on the same inputs?

**Reproducibility:**

3: Could reproduce the results with some difficulty. The settings of parameters are underspecified or subjectively determined; the training/evaluation data are not widely available.

**Reviewer Confidence:**

4: Quite sure. I tried to check the important points carefully. It's unlikely, though conceivable, that I missed something that should affect my ratings.

---

> ### Author Rebuttal · Authors · 2023-08-28
>
> Thank you for your detailed feedback on our paper. We appreciate your comments on the method's flexibility and effectiveness.
>
> Regarding the novelty, we assure you that our approach stands apart:
>
> (1) We employ an "open" search space guided by the LLM itself, differing from TEMPURA and GRIPS whose search space is limited to a discrete set of predefined phrase-level operations (delete, swap, paraphrase, add)
> (2) Our method is nonparametric and only requires LLM API access (unlike RLPrompt and TEMPURA).
> (3) Using textual "gradients" to guide text optimization in a directed way and bypass the discrete optimization barrier is a significantly novel algorithmic technique at the time of submission.
> (4) We offer the first comparative evaluation of bandit algorithms in a prompt optimization setting.
>
> We apologize for the confusion when introducing GRIPS and TEMPURA together. You're totally right that GRIPS isn't an RL algorithm. We introduced them together because GRIPS and TEMPURA employ a similar search space over prompts (the same phrase-level operations: delete, swap, paraphrase, add), and so GRIPS has similar fundamental characteristics as TEMPURA (do you want to choose from a pool of edit operations using an RL-trained orchestrator (TEMPURA), or enumerate-and-select based on end task performance + prediction entropy (GRIPS) ). Since we compared directly against TEMPURA (our "RL" baseline), this baseline suggested an upper bound on GRIPS performance as TEMPURA selects from similar operations with more efficiency.
>
> On the topic of RLPrompt, our paper focuses on both gradient-free and nonparametric optimization algorithms that produce comprehensible outputs using LLM API only, making direct comparison with RLPrompt less relevant. This is because RLPrompt requires low-level access to all models, trains a block of parameters in the prompt-generating LLM and produces incomprehensible outputs.
>
> In response to your questions:
>
> A) We've run a direct comparison against GRIPS with 25 evaluations per candidate, confirming it underperforms compared to TEMPURA and ProTeGi. The full results will be in the camera-ready version: https://imgur.com/ha1IIMX
>
>
> B) The performance drop is due to the variance in prompt generation and bandit selection; however, the Y-scale on those graphs is relatively small, suggesting minor fluctuations.
>
>
> Qualitative comparisons against previous work (including TEMPURA ("RL" baseline) and GRIPS search space) can be found in table 4 and the Appendix. We will include more detailed qualitative studies in the camera-ready version.
>
> Thank you for bringing these issues to our attention. We're eager to improve the paper based on your suggestions.

---

### Meta-Review · Area_Chair_j3QZ · 2023-09-18

**Recommendation:** 4

**Metareview:**

This paper proposes a chain-of-thought style prompt editing method that iteratively enhance initial prompts with techniques of beam search and bandit algorithms. It demonstrates its efficacy on several classification benchmark datasets. Experiments are well-executed and comprehensive baselines are well compared. However, the proposed method requires minibatches of a training set and large amounts of queries, which seems to be inference-expensive for practical setups.

---

### Decision · Program_Chairs · 2023-10-07

**Decision:**

Accept-Main

**Comment:**

This paper proposes a chain-of-thought style prompt editing method that iteratively enhance initial prompts with techniques of beam search and bandit algorithms. It demonstrates its efficacy on several classification benchmark datasets. Experiments are well-executed and comprehensive baselines are well compared. However, the proposed method requires minibatches of a training set and large amounts of queries, which seems to be inference-expensive for practical setups.